# Effects of behavioural parent training for children with attention-deficit/hyperactivity disorder on parenting behaviour: a protocol for an individual participant data meta-analysis

Laura Steenhuis  ,[1] Annabeth P Groenman  ,[1] Pieter J Hoekstra,[1] Rianne Hornstra,[1] Marjolein Luman,[2,3] Saskia van der Oord,[4] Barbara J van den Hoofdakker[1]

LS and APG contributed equally.

LS and APG are joint first authors.

For numbered affiliations see end of article.

**Correspondence to**
Dr Annabeth P Groenman;
a.groenman@accare.nl

## ABSTRACT

**Introduction** Behavioural parent training (BPT) is a well-established treatment for children with attention-deficit/hyperactivity disorder (ADHD). BPT is based on the hypothesis that improvements in parenting are mediators of improvements in children's behaviours. However, meta-analyses show considerable heterogeneity in effects of BPT on child outcomes, and meta-analyses on parenting outcomes are scarce. Also, few studies have investigated parenting factors as mediators of child outcomes. This study aims to examine the effects and moderators of BPT on parenting outcomes and whether improvements in parenting mediate amelioration of behaviour and impairment in children with ADHD.

**Methods and analyses** We will conduct an individual participant data meta-analysis (IPDMA), making use of individual data of existing trials, and giving the opportunity for highly powered moderator analyses. This IPDMA will be performed by the Psychosocial ADHD INTervention (PAINT) collaboration. We will include randomised controlled trials of BPT, for individuals with ADHD below 18 years old. Systematic searches have been performed to locate relevant papers. Authors are currently contacted to share their data with the PAINT-IPDMA project. We will examine effects of BPT on parenting outcomes (eg, positive and negative parenting, management of affect, perceived parenting competence, parenting stress), moderators of these effects (eg, parental depression, parenting stress, severity of the child's ADHD symptoms) and subsequently perform mediation analyses where parenting outcomes are modelled as mediators of child outcomes (eg, symptoms and severity of ADHD, comorbid psychopathology and impairment).

**Ethics and dissemination** We will include data from randomised control trials for which ethical approval has been received and consent forms have been signed. Deidentified data will be provided by the original investigators. We aim to disseminate our findings through peer-reviewed scientific journals, presentations at (inter) national scientific meetings, newsletters, the website of our project and the Dutch academic workspace ADHD.

**PROSPERO registration number** CRD42017069877.

### Strengths and limitations of this study

► To our knowledge this is the first individual participant data meta-analysis (IPDMA) that examines the effects of behavioural parent training for attention-deficit/hyperactivity disorder on parenting outcomes.
► By using IPDMA, we are able to conduct highly powered moderation analysis.
► By collecting individual participant data from randomised controlled trials, we have the opportunity to perform uniform analyses across studies.
► Our IPDMA includes only variables that are reported consistently across the majority of studies in the database.
► Representativeness of the database will depend partly on the willingness and ability of investigators to share their data.

## INTRODUCTION

Behavioural parent training (BPT) is a well-established intervention for children with attention-deficit/hyperactivity disorder (ADHD) and recommended as first line treatment by many international guidelines.[1–3] In BPT, parents are trained to apply behavioural techniques meant to increase prosocial and adaptive child behaviours and to reduce disruptive and maladaptive child behaviours. In addition, BPT focusses on nurturing and positive parent-child relationships.[4] Several meta-analyses on BPT for children with ADHD have shown medium to large effect sizes (ES) on parent-reported reductions of ADHD symptoms, comorbid problems, impairment and parenting behaviours.[3 5–7] However, the full range of relevant parenting outcomes of BPT has not been assessed in meta-analyses.[7 8] Moreover, while improvements in parenting behaviours are thought

to mediate improvements in children's behaviours and associated impairments,[3 9] studies investigating these mediation effects are scarce. Another unresolved issue is that individual studies commonly lack the statistical power to adequately assess moderators of improvements of parenting outcomes. Knowing which moderators impact the effectiveness of BPT for child[5] and parenting outcomes,[3 6] in addition to knowing which mediators drive the effects of BPT will give more insight into for whom BPT works best and into the working mechanisms underlying this, therefore contributing to personalised treatment of children with ADHD.

BPT aims to increase the demonstration of positive parenting behaviours and to inhibit negative parenting behaviours, improve the management of affect, reduce parenting stress and enhance parenting self-efficacy,[9] in order to positively influence children's behaviours and decrease impairment. Parents are trained to modify environmental antecedents (eg, providing structure) and consequences (eg, positive rewards for adaptive behaviour) of behaviour. Positive parenting behaviours consist of providing praise, encouragement, effective communication, demonstrating positive effect and offering physical affirmations,[10 11] whereas negative parenting behaviours are described as providing inconsistent discipline, verbal criticism, corporal punishment, poor supervision and demonstrating negative effect.[12–14] Essential to positive parenting behaviour is the effective management of parental affect, which is often a specific target of BPT.[15] Parents are taught to express positive emotions (eg, love, affection and warmth) and to inhibit expressions of negative and unsupportive emotions (eg, anger, frustration and irritability).[16] Parenting stress (ie, stress arising from the feeling that the demands of parenting outweigh the resources[17]) is often reported by parents of children with ADHD[18] and is also an important target of BPT. Lastly, parenting self-efficacy is an important target of BPT[19]; parents should feel more confident and competent in carrying out their parenting tasks.

When assessing parenting outcomes of BPT, it is important to consider a range of outcomes, particularly positive and negative parenting behaviours, management of affect, parenting stress and parenting self-efficacy. However, treatment trials of BPT have not uniformly assessed all parenting outcomes that are explicit targets of BPT,[3 6] although BPT outcomes may differ for these different domains. For example, a recent systematic review investigated multiple parenting outcomes (display of parental affect, parenting stress and parenting self-efficacy) of BPT and showed positive results for outcomes most closest to the target of BPT (eg, parenting self-efficacy), but less so for more distal outcomes (eg, parental psychopathology).[20] Notably, studies in that review were not specifically conducted in ADHD samples, nor were outcomes quantified using a (individual participant data) meta-analytic approach, which will be the analytic approach in the current study. Recently, two meta-analyses examined the effectiveness of behavioural interventions

(mostly consisting of BPT) on parenting outcomes in samples of children with ADHD, either assessed by raters unblinded to the treatment condition (eg, parents involved in BPT[9]) or by blinded raters (eg, independent raters coding video-taped interactions between the parent and child[3 6]). The first meta-analysis[3] considered effects on positive and negative parenting behaviours and parenting self-efficacy, immediately postintervention. Results indicated improvements in positive parenting behaviours (medium ES of 0.68 and 0.63 for unblinded and blinded raters, respectively), reductions in negative parenting behaviours (medium to small ES of 0.57 and 0.43 for unblinded and for blinded raters, respectively) and improvements in parenting self-efficacy (small ES of 0.37 for unblinded raters). The other meta-analysis[6] investigated the effects of BPT for preschool children with (or at risk for) ADHD, and reported medium to small ES of 0.63 and 0.33 by unblinded and blinded raters, respectively, for reductions in negative parenting behaviours. In addition, long-term effects (up to 12 months) on negative parenting behaviours were reported by unblinded raters with an ES of 0.12 (very small). Overall, both unblinded and blinded parent outcomes show improvements after BPT, although the effects on blinded outcomes generally appeared somewhat smaller than the effects on unblinded outcomes. Improvements on other relevant domains such as parenting stress and display of parental affect have not been addressed in meta-analyses, although individual studies demonstrated that parenting stress can be effectively alleviated with BPT,[19] and that mothers who participated in BPT have less negative effect and better emotion regulation abilities post-treatment.[21]

Heterogeneity in ES is common in meta-analyses investigating parenting outcomes of BPT for children with ADHD.[3 6] More knowledge on factors associated with treatment effects will yield more insight into for whom BPT works best, and may allow clinicians to make better treatment choices tailored to individuals. So far, very few randomised control trials (RCTs) were adequately powered for moderator analyses, and existing moderator analyses were mostly limited to child behavioural outcomes.[22–24] While traditional methods of addressing heterogeneity in meta-analyses (such as excluding extreme ES, subgroup analyses or meta-regression) may resolve heterogeneity (ie, reduce $I^2$), they reveal little about the cause of this heterogeneity. Individual participant data meta-analysis (IPDMA) includes data at an individual level rather than at study level, which enables exploration of moderators, therefore yielding more information about the cause of heterogeneity. Moreover, IPDMA performs a uniform analysis across all studies. IPDMA also has enough power to perform subgroup analyses, which most individual RCTs lack. Due to the collaborative nature of an IPDMA, collaborators can provide input on all phases of the research (including design, analyses, interpretation and manuscript preparation), leading to a high-quality product.[25–27] So far, no IPDMA has been conducted for parenting outcomes of BPT for children with ADHD.

In this paper, we present our protocol for an IPDMA on BPT for children with ADHD. The current IPDMA will explore several child-related and parent-related moderators on parenting outcomes of BPT. Given that the examination of moderators on parenting outcomes is scarce, the choice of possible moderators to be investigated in this IPDMA will also be drawn from BPT trials on child outcomes. First, age of the child might be an important moderator,[28] as parents may have more influence on younger children and younger children may have less engrained symptoms. A meta-analysis indeed demonstrated that BPT had more effect on positive parenting behaviours for younger compared with older children with ADHD.[3] Second, medication use of the child may moderate BPT outcomes, as parents of children on medication might find it easier to adapt their parenting styles and experience less resistance when doing so, although there could also be a floor effect as that children with medication might already function better. Previous studies suggested that medication use may positively contribute to BPT outcomes on child's ADHD symptoms,[29] although results have been mixed.[22] Third, it seems plausible that intelligence of the child is positively associated with a treatment that involves learning associations between behaviours and consequences and new skills.[30] There is some evidence that higher children's Intelligence Quotient (IQ) is associated with more improvement in ADHD symptoms following behavioural treatment, but only for specific subgroups of children, such as girls with more antisocial symptoms.[24 31 32] Fourth, pretreatment ADHD severity and presence of comorbidities could moderate BPT, as parents might find it easier to change their parenting behaviour when the child has less complex symptomatology. Previous individual studies have indeed confirmed that comorbidities at baseline negatively impacted the outcome of BPT treatment with regard to child symptoms.[22 23 33]

There are several parent-related moderators that may have an effect on parenting outcomes of BPT. First, parental mental health problems (depression, ADHD, parental stress) are likely to affect how well parents are able to grasp new information and impact their ability to learn new methods of parenting behaviours and skills.[22 23 34–36] Second, low socioeconomic status (SES) may moderate outcomes of BPT, as this may increase family strain and impact the availability of family resources. There is mixed evidence of the moderating effect of low SES on the effects of BPT on ADHD symptoms and related problems, showing either no effect,[24] or better outcomes for parents with lower SES.[37] So far, effects of low SES in relation to parenting outcomes of BPT for children with ADHD have not been investigated. For the present IPDMA we will rely on an imperfect measure of SES, that is, parental education, as we are reliant on the measures that are consistently used across studies Third, single parenthood may also moderate outcomes of BPT, as individual studies have demonstrated that single mothers are less likely to respond to BPT,[38 39] and maintain treatment gains over time[40] in terms of parenting behaviour. Indeed, research suggests that single parents benefit more from enhanced BPT programmes, suited to their specific needs,[41 42] compared with standard BPT programmes. Fourth, there is evidence that lower parenting self-efficacy at baseline has a negative impact on improvements in behavioural problems in children with ADHD following BPT[43] and one could expect similar results for parenting outcomes. On the other hand, it is also possible that lower parenting abilities at baseline yield more room for improvement in BPT, and thus both directions of the effect could be expected.

Following the investigation of parenting outcomes of BPT as part of our IPDMA, the next step will be to examine mediators of improvements in child's behaviour following BPT. Currently, there are no meta-analyses analysing whether improvements in parenting behaviour mediate improvements of BPT on symptoms of ADHD of the child. Some evidence of individual studies shows that reductions in parenting behaviours or parenting attributions mediate improvements on child outcomes in behavioural interventions.[12 35 44] This IPDMA aims to synthesise the available data regarding the association between improvements in parent outcomes and improvements in child's outcomes following BPT.

The specific aims of our IPDMA include:
1. To investigate effects of BPT on parenting outcomes (positive and negative parenting behaviours, display of parental affect, parenting stress and parenting self-efficacy). Given that BPT for ADHD has shown different effects when assessed by unblinded raters and by more blinded raters,[3 6 7] we aim to distinguish (if possible) between unblinded and blinded assessments of parenting behaviour and display of parental affect.
2. To investigate possible child (including age, medication use, IQ, ADHD severity, presence and severity of comorbidities) and parent (including depression, ADHD, SES, single parenthood and parenting measures) moderators of parenting outcomes.
3. To investigate whether improvements in parenting behaviours mediate the effect of BPT on behavioural child outcomes and impairment following treatment.

For aims 1–3, outcomes of BPT will be examined immediately post-treatment and at long-term where possible.

## METHODS AND ANALYSIS

For this IPDMA we will build on the Psychosocial ADHD INTervention IPD (PAINT-IPD) database which is registered in PROSPERO (https://www.crd.york.ac.uk/PROSPERO/display_record.php?ID=CRD42017069877&ID=CRD42017069877). In this project, we collect data on psychosocial treatments for children with ADHD: the search is regularly updated and the database continues to expand. For the purpose of the current IPDMA on parenting outcomes, we will use the same methods and search strategy. This protocol is written in line with the Preferred Reporting Items for Systematic Reviews and

Meta-Analysis-Protocols (PRISMA-P) 2015 checklist[45] (see online supplemental checklist). The current study is planned to commence in September 2020.

## Inclusion criteria

We will include RCTs of behavioural treatments of individuals aged below 18 with ADHD, corroborated by clinical cutoffs on questionnaires or (semi)-structured interviews. We will include studies that compared BPT with a control condition (ie, all conditions that are labelled control, including active treatment), and studies that compared BPT to another behavioural intervention (head to head comparisons). We excluded studies or intervention arms that used optimised medication treatment next to BPT as part of their study design or as a control condition. We define BPT as interventions directed at changing children's behaviours (ie, increasing desirable behaviours and decreasing undesirable behaviours), using (cognitive) behavioural therapeutic techniques which parents are trained in.[46] Multimodal interventions (consisting of both parent and/or teacher and/or child training), will be included if the time spent on parent training within the intervention was at least equal to other types of training.

## Selection and screening of studies

The last systematic search was performed on 13 May 2020. Currently, we are contacting the authors of the newly identified studies. Two authors (APG and RH) performed the selection and screening of studies, disagreement was resolved by consensus. A two-step approach to identifying relevant articles was used. First, Medline, CINAHL, PsycINFO, EMBASE+EMBASE CLASSIC, ERIC, Web of Science (Science Citation Index Expanded) was searched for relevant papers using a combination of the following search terms and their synonyms, as well as hierarchical family form (eg, MeSH terms): treatment-specific terms (eg, behavioural treatment, psychosocial treatment and parent training), ADHD, child and RCT. No date restrictions were applied. English, German and Dutch language publications published in peer-reviewed journals were included. Second, literature lists of all selected studies and relevant systematic reviews and meta-analyses were handsearched to identify possible missing articles (complete search criteria for each database are available in online supplemental 1.

## Data collection and management
### Author contact

We will contact the corresponding authors of all eligible trials to ask for their participation in the current IPD meta-analyses. If after several weeks we have had no response, we will send a reminder to the corresponding author. If we have failed to establish contact with the corresponding author, we will email the other authors of the study. Furthermore, we will contact researchers during conferences, and through our personal network to retrieve all eligible databases.

## Data format and management

Part of the data has already been received in light of our previous IPDMA.[5] For newly identified data that we have not yet received, we will use the best, safest way to transfer the data. The most convenient way for most authors will be to transfer the encrypted data per email to the project management, but we will be open to other options (eg, face to face transfer). A copy of our data collection manual for the PAINT-IPD database can be found in online supplemental 2. We will allow authors to send the data in all possible formats, although the most preferable format would be one in which each subject represents a row and each variable a column.

If authors are not included in the PAINT collaborators group yet,[5] we will offer one or two authors of each included study a place in our PAINT collaborators group. We plan yearly telephone calls and/or meetings at large international conferences to keep the working group up to date and discuss design and methodological issues. Members of the working group who have provided data on BPT trials with parenting outcomes will have the opportunity to provide feedback on the first draft of the manuscript and will be sent a copy of the final manuscript before submission. Only authors in the IPD steering committee group (APG, BJvdH, SvdO, ML and PJH) will have access to the data.

## Ethics and dissemination

The original investigators will be asked for de-identified data, so that only the original investigator knows the link between data and participant. We will only include RCTs where ethical approval has been given and participants signed consent forms. Since the current IPDMA is an extension of the original purpose of the eligible studies, we do not expect any ethical issues with the current IPDMA. Results of our study will be disseminated through peer-reviewed scientific journals, and presentations on (inter)national scientific and/or clinical expert meetings.

## Patient and public involvement

An expert panel consisting of parents of children with ADHD was organised at the conception of the idea in which feedback was given to the plans and changes were made accordingly. Our results will be communicated to clinicians, clients and their parents through newsletters and through the academic workplace 'ADHD en druk gedrag' (also see https://adhdendrukgedrag.nl/) in which many parents, client organisations and mental healthcare professionals are represented.

## Variables

For the PAINT-IPD database, a data request form will be sent to all authors of the original studies, containing a list of variables that will be requested (see online supplemental 2a). This list was determined by reviewing the literature and the IPDMA steering committee (authors APG, SvdO, ML, PJH and BJvdH) assessed these outcomes

domains for suitability and interest. The final list of variables will depend on the available data of all studies.

For the current protocol, the following parenting variables will be selected:

### Parenting variables

*Positive and negative parenting behaviours* will be assessed using unblinded and blinded measures. For unblinded parenting measures, we will select questionnaires such as the Alabama Parenting Questionnaire.[10] For blinded measures, we will select assessments used to observe and code parent behaviour, such as the Dyadic Parent-Child Interaction Coding System.[47]

*Display of positive and negative parental affect* will be assessed with measures specifically designed to capture parental affect (eg, 5-minute speech sample of expressed emotion[15];) or a subscale of an existing parenting scale (eg, attachment domain of the Parenting-Relationship Questionnaire[48]).

*Parental stress* will be assessed using measures specifically designed to capture stress from parenting, such as the Parenting Stress Index[49], but also measures assessing stress in the caregiver more generally (Depression and Anxiety Stress Scales (DASS)[50]).

*Parenting self-efficacy* will be assessed using measures specifically designed to capture parenting self-efficacy, such as the Parental Sense of Competence Scale,[51] a subscale of an existing parenting scale (eg, the Parenting-Relationship Questionnaire[48]) or a scale designed to assess parenting competence in specific contexts, such as education (eg, Parent as Educator Scale[52]).

*Parental depression* will be assessed using measures to capture depressive symptoms, such as the Beck Depression Inventory[53], a subscale of an existing mental health questionnaire (eg, depression subscale of the DASS[54]) or a mental health questionnaire for which the overall score can be used as a proxy for depression (eg, the General Health Questionnaire[55]).

*Parental ADHD* will be assessed using an adult measure of ADHD, such as the Adult Self-Report Scale Screener.[56]

The remaining parent variables, including *single parenthood* and *SES,* are commonly assessed with demographic questionnaires or items. SES can be assessed in different ways (eg, based on income, occupation, neighbourhood or education) and due to our IPDMA methodology, we are reliant on choosing the assessment measure that is most consistently used across studies. Given that data on parental education is often available, and is deemed acceptable as a proxy for SES,[5 57] parental education level will serve as a proxy measure for SES in the current study. SES will be conceptualised as low (<high school), medium (high school graduate or college education) or high (>college graduate).

Note that *parenting behaviours, parental affect, parental stress* and *parenting self-efficacy* will serve as moderators (to predict parenting outcomes), mediators (in the relationship between BPT and child outcomes) and outcome variables (in the moderator analysis). *Parental depression, parental ADHD, single parenthood* and *SES* will serve as baseline moderators to predict parenting outcomes.

### Child variables

Additionally, the following child variables will be selected:

*Child ADHD severity* will be assessed using a parent-rated measure of childhood ADHD, such as the ADHD subscale of the Connors Parent Rating Scale.[57]

*Comorbidity* will be assessed using symptoms of oppositional defiant disorder (ODD) and/or conduct disorder (CD), by for example the ODD subscale of the Connors Parent Rating Scale,[57] or the CD subscale of the Disruptive Behaviour Disorder Rating Scale.[58] In addition, internalising symptoms will be assessed, using for example the internalising subscale of the Child Behaviour Checklist.[59]

*Global impairment* will be assessed using a parent-rated or clinician-rated measure of global impairment, such as the Impairment Rating Scale.[60]

The remaining child variables, including *child age, medication use and IQ* are commonly assessed with demographic questionnaires or items.

*Child ADHD severity, comorbidity and global impairment* will serve as moderators (to predict parenting outcomes) and outcome variables (in the mediation analysis). *Child age, medication use and IQ* will solely serve as moderators to predict parenting outcomes.

For all variables (moderators and outcomes), when a study reports multiple measures to capture the same concept, the measure which is most often used by other studies will be included. If multiple raters (mothers and fathers) are included to assess the same concept, the assessment by the mother will be preferred as they are more often the primary caregiver and more often take part in BPT. Regarding harmonisation, for each dataset continuous measures will be converted into z-scores, using pre-intervention-score SD within studies.

### Quality assessment

Quality assessment will be done independently by three authors (a combination of APG, RH and LS) using Cochrane risk of bias. Any disagreement will be resolved by consensus. Once the data has been received, all raw data sets will be checked for impossible, missing or extreme values. We will collect data on all randomised participants. As this will possibly reintroduce participants who were previously excluded, we will also check randomisation parameters (eg, age, sex and ADHD severity of the participants). If any unexpected deviations are found between our results and the published results, the original researcher will be contacted to locate the origin of this deviation.

### Analysis

Effects of BPT will be calculated using a one-stage IPDMA, in which data from participants across studies will be analysed in one stage, clustered by study. A linear multi-level analysis will be used to examine the effects of BPT on parenting outcome measures. A random intercept

for study will be added to each model. Post intervention outcome measures will be used as dependent variables in these models, and preintervention outcome measures and intervention group will be added. The interaction between intervention group and moderators of interest will be added to the model to assess their moderating effect on treatment outcomes. To analyse changes in parenting behaviours as a potential mediator in the relationship between BPT and child outcomes, the change in parenting behaviours (post-treatment score – baseline score) will be used. An interaction between intervention group and change in parenting behaviours will be added to the model to determine whether change in parenting behaviours had a main effect on child outcomes and/or an interactive effective with treatment.[61 62] We will conduct sensitivity analyses between studies that provided data and those that did not, for demographic characteristics (eg, age) and for inclusion criteria (cut-off on measures vs meeting diagnostic criteria for ADHD) and reported ES. If sufficient data is available, a subgroup analysis for child age will be conducted, where all analyses will be repeated separately for pre-school children (<6 years of age), school-aged children (6–12 years of age) and adolescents (>12 years of age).

## DISCUSSION

The current protocol presents the first IPDMA to synthesise research findings on treatment effects of BPT for children with ADHD on parenting outcomes. Both child and parent moderators of parenting outcomes will be explored and a mediation analysis will be conducted to examine whether changes in parenting behaviours mediate the effect of BPT on child behaviours and impairment. Previous efforts to identify moderators of parenting outcomes are scarce, as there are few well-powered RCTs and only two meta-analyses on this topic.[3 6] The heterogeneity found in parenting outcomes in previous meta-analyses suggests that there is not a 'one size fits all' solution of BPT for children with ADHD. Parents and children who differ with regard to their personal, clinical and demographic characteristics, are also likely to differ in their response to BPT. Clearly, an IPDMA approach is needed to further examine the effects of BPT on parenting outcomes, to elucidate potential sources of heterogeneity among children and parents, and to investigate potential mechanisms of change in BPT for children with ADHD. Ultimately, both research and clinical practice may be informed by the knowledge of which child and/or parent responds best to a certain treatment, thereby contributing to the overall goal of providing the best care to children with ADHD and their parents.

An additional important aim of the current IPDMA will be to examine a crucial assumption about the working mechanism of BPT, namely whether improvements in parenting behaviours mediate improvements in child behaviour and impairment. So far, single studies have examined this assumption, finding some supportive

evidence,[12 19] but this hypothesis has never been addressed in a meta-analysis, let alone by using an IPDMA approach. By synthesising all available (raw) data from RCTs, we will conduct highly sophisticated and powered statistical analyses, to provide insight into the working mechanisms of BPT. This knowledge will allow us to further improve and refine BPT programmes for children with ADHD.

Despite clear strengths of the IPDMA approach, such as high-powered moderator, mediation and subgroup analyses, there are some limitations as well.[25 27 63] First, IPDMA cannot change anything about the way the study was originally conducted. Second, it is unclear whether enough studies measured the variables of interest, and thus whether all intended outcomes can be examined. Third, if investigators of trials are unable to share their data (or may not be willing to share their data), not all relevant data can be included in the synthesis. Fourth, not all factors which may affect study outcomes can be conceptualised as a moderator, such as efficacy versus effectiveness trials (ie, studies often do not fit exactly into one of both categories). Fifth, since parental education levels are easy to collect, they are omnipresent in the data. Ideally, several aspects of SES are taken into account, such as income, occupation, and/or neighbourhood,[64 65] but this data is available in only a few datasets.

Notwithstanding these limitations, this study has the potential to elucidate clinically relevant questions concerning the efficacy of BPT for children with ADHD and their parents and to provide insight in moderators and mediators of treatment effects. This knowledge may improve and optimise current treatment programmes and could eventually lead to advances in personalised treatment for children with ADHD.

**Author affiliations**
[1]Department of Child and Adolescent Psychiatry, University of Groningen, University Medical Center Groningen, Groningen, the Netherlands
[2]Dept. Clinical Neuropsychology, Vrije Universiteit Amsterdam, Amsterdam, The Netherlands
[3]Bascule, academic centre for child and adolescent psychiatry, Amsterdam, the Netherlands
[4]Clinical Psychology, KU Leuven, Leuven, Flanders, Belgium

**Acknowledgements** The authors would like to thank all collaborators of the PAINT-IPDMA project for providing input to this protocol and for providing data for this project. The authors thank our parents of clients' expert panel for their valuable input during the conception of the idea for the PAINT-IPDMA.

**Contributors** BJvdH is the guarantor. APG wrote the first draft of the manuscript, and LS wrote a revised and final version. APG, RH and LS are performing the systematic search and data extraction. PJH, ML, SvdO and BJvdH acquired funding for the PAINT-IPDMA project; an international database including individual participant data from RCTs examining psychosocial interventions for the treatment of ADHD in children. The current study is a part of the PAINT-IPDMA project. PJH, ML, SvdO, BJvdH and APG setup the initial design for the IPDMA. All authors worked together to design this IPDMA, contributed to the intellectual content of the current manuscript and approved the final version.

**Funding** This research was funded by The Dutch Organization for Health Research and Development (ZonMw) under grant number 729300013.

**Disclaimer** The funder had no role in the design of this protocol. The funder will have no input on the collection of data, the data analysis or the interpretation or publication of the study results.

**Competing interests** None declared.

**Patient consent for publication** Not required.

**Provenance and peer review** Not commissioned; externally peer reviewed.

**ORCID iDs**
Laura Steenhuis http://orcid.org/0000-0002-1256-3754
Annabeth P Groenman http://orcid.org/0000-0002-8394-6605

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
