## [Reviewer comments · BMJ Open]

ARTICLE DETAILS

TITLE (PROVISIONAL)	Effects of Behavioral Parent Training for children with attention-deficit/hyperactivity disorder on Parenting Behavior: A protocol for an Individual Participant Data Meta-analysis
AUTHORS	Steenhuis, Laura; Groenman, Annabeth; Hoekstra, Pieter J; Hornstra, Rianne; Luman, Marjolein; van der Oord, Saskia; van den Hoofdakker, Barbara

VERSION 1 – REVIEW

REVIEWER	Anil Chacko New York University, USA
REVIEW RETURNED	14-Mar-2020

GENERAL COMMENTS	This is an important paper that has the potential to inform key questions as it relates to BPT for ADHD. I commend the authors for the thoughtfulness of their approach and the efforts toward executing the study. I provide a few areas for consideration: 1) It is unclear whether it is appropriate to include adolescents in a study of BPT. No doubt that there are behavioral interventions for adolescents with ADHD but these are qualitatively different than what traditional BPT is (in both content, format and process). I agree with the authors that assessing the moderating role of child age is important, but I would consider this a comparison between preschool versus school-age children.2) The study includes youth with ADHD diagnosed by various means (cut-offs on measures as well as those who meet diagnostic criteria). These may be two distinct groups and “approach” to classification may moderate outcomes as well.3) SES and single parent status are important factors, as a limitation of the ADHD literature is on the inclusion of studies that represent the full range of key demographic variables (Evans et al., 2018). I do wonder if the experience of parents who may be from lower SES or single-parents may vary as a function of the context (i.e., country) in which they live. SES and single parenthood is experienced differently when there are more robust “social-safety-nets” to support parents.4) Given work on the potential differences between effectiveness and efficacy samples, this may be an important moderating factor to consider.5) For the text on single parenthood, I would cite the work of Chacko and colleagues as the work currently cited is not specially for ADHD. Chacko, A., Wymbs, B.T., Arnold, F.W., Pelham, W.E., Swanger-Gagne, M., Girio, E.L., Pivics, L., Herbst, L., Guzzo, J.L., Phillips, C., & O'Connor, B. (2009). Enhancing traditional behavioral parent
--

	training for single-mothers of children with ADHD. Journal of Clinical Child and Adolescent Psychology, 38, 206-218. Chacko, A., Uderman, J.* , & Zwilling, A.* (2013). Lessons learned in enhancing behavioral parent training for high-risk families of youth with ADHD. The ADHD Report, 6-16. 6) Page 5, line 48-52 states” Parents are taught to express positive emotions (e.g., love, affection and warmth) and to inhibit expressions of negative and unsupportive emotions (e.g., anger, frustration and irritability) [16].” Is it more accurate to say that parents are to express positive behaviors and inhibit negative behaviors? You may be irritated (that’s a feeling/emotion) but it’s the behavioral expression that is problematic. That is true for positive emotions- these are likely better framed as behaviors to exhibit (from a BPT perspective). 7) The authors should review the Colalillo and Johnston meta-analysis as its relevant to these issues of the proposed work: Colalillo, S., & Johnston, C. (2016). Parenting Cognition and Affective Outcomes Following Parent Management Training: A Systematic Review. Clin Child Fam Psychol Rev, 19(3), 216-235. doi:10.1007/s10567-016-0208-z
--	---

REVIEWER	Nicole K Schatz Florida International University
REVIEW RETURNED	14-Apr-2020

GENERAL COMMENTS	Overall, this protocol is clear and methods appear consistent with best practices for meta-analyses. A minor note, there are a couple areas that might benefit from additional explanation. 1. The first is with regard to socio-economic status. SES is reported across studies using a variety of metrics. It would be helpful if the authors could provide additional description of how SES will be defined for the purposes of this meta-analysis. 2. On page 14, the authors provide helpful description of how multiple measures of the same concept will be addressed in the analyses. It would be helpful if this explanation could be expanded to include a description of how multiple raters will be addressed. For example, how will the authors address studies in which ratings are provided by both mothers and fathers?
---

VERSION 1 – AUTHOR RESPONSE

Reviewer: 1

Reviewer Name: Anil Chacko

Institution and Country: New York University, USA

Please state any competing interests or state ‘None declared’: None declared

Please leave your comments for the authors below

This is an important paper that has the potential to inform key questions as it relates to BPT for ADHD. I commend the authors for the thoughtfulness of their approach and the efforts toward executing the study.

We thank the reviewer for the positive remarks and appreciation of our paper.

I provide a few areas for consideration:

1) It is unclear whether it is appropriate to include adolescents in a study of BPT. No doubt that there are behavioral interventions for adolescents with ADHD but these are qualitatively different than what traditional BPT is (in both content, format and process). I agree with the authors that assessing the moderating role of child age is important, but I would consider this a comparison between preschool versus school-age children.

We agree with the reviewer that BPT is primarily directed at parents of preschool and school-aged children with ADHD. On the other hand, BPT is also be applied to parents of adolescents, though this is less frequently the case. Given that most studies focus on preschool and school aged children with ADHD, we have removed the term 'adolescents' from our title.

The aim of our study is to investigate parent-mediated treatment in ADHD (regardless of the child's age), and therefore we feel that adolescent studies that meet our definition should be included. This is especially important, as for all age groups (preschool children (<6 years), school aged children (6-12 years) and adolescents (<12 years)), the techniques will differ slightly. We feel that there is information in this difference and therefore we have decided to perform subgroup analyses (if the data allows us) where we will repeat all analyses in those specific subgroups. We have added this information to the analysis section, see page 14: "If sufficient data is available, a subgroup analysis for child age will be conducted, where all analyses will be repeated separately for pre-school children (<6 years of age), school aged children (6-12 years of age) and adolescents (>12 years of age)."

2) The study includes youth with ADHD diagnosed by various means (cut-offs on measures as well as those who meet diagnostic criteria). These may be two distinct groups and "approach" to classification may moderate outcomes as well.

We thank the reviewer for this useful suggestion. Including both groups of studies may be of importance given that there is an increasing number of studies showing that children with clinical ADHD symptoms, without necessarily meeting diagnostic criteria for ADHD, show similar functional impairments and may profit similarly from behavioural interventions compared to children meeting diagnostic criteria for ADHD (e.g., Kirova et al., 2019). Currently, although we have this broad inclusion criterion, none of the studies that were identified in the search actually only used a clinical cut-off on a questionnaire to determine inclusion into the trial. If we will include studies that use these cut-offs in the future, we will conduct sensitivity analyses without these studies. We have added this to the protocol on page 13:

"We will conduct sensitivity analyses between studies that provided data and those that did not, for demographic characteristics (e.g., age), and for inclusion criteria (cut-off on measures versus meeting diagnostic criteria for ADHD)".

3) SES and single parent status are important factors, as a limitation of the ADHD literature is on the inclusion of studies that represent the full range of key demographic variables (Evans et al., 2018). I do wonder if the experience of parents who may be from lower SES or single-parents may vary as a function of the context (i.e., country) in which they live. SES and single parenthood is experienced differently when there are more robust "social-safety-nets" to support parents.

We think the reviewer has raised an interesting and important point. Nonetheless, the aim of the current study is to use IPDMA in order to assess individual characteristics of the parent (e.g., depression severity) or the child (e.g., IQ) (factors which vary on an individual level within studies themselves). Given that we are interested in examining specific and individual characteristics of both the parent and child, and doing so will make use of the full strength of the IPDMA approach, we feel

the proposed factor 'context' is out of the scope of the current study. Another consideration we have is that the majority of studies to date are performed in western countries (US, Europe) and we feel that, unfortunately, there will be too little variation in countries (and thus cultural, social and societal characteristics) to take this into account.

4) Given work on the potential differences between effectiveness and efficacy samples, this may be an important moderating factor to consider.

We agree that distinguishing between effectiveness and efficacy trials is important, and could be a factor to consider in our analysis. The problem is that in the studies that we included so far, the "pure" definitions of effectiveness and efficacy are not well represented. For example: some have well defined samples with strict diagnostic criteria but were conducted in multiple clinical settings with less control on internal validity. It would therefore be difficult to classify studies into one category (effectiveness) or the other (efficacy), or to determine a single variable that would be a proxy for this (although we are open for suggestions). This is why we feel that it is not possible for us to analyse this in the current dataset. Therefore, we have added this limitation to our discussion section, see page 14:

"Fourth, not all factors which may affect study outcomes can be conceptualized as a moderator, such as efficacy versus effectiveness trials (i.e., studies often do not fit exactly into one of both categories)."

5) For the text on single parenthood, I would cite the work of Chacko and colleagues as the work currently cited is not specially for ADHD.

Chacko, A., Wymbs, B.T., Arnold, F.W., Pelham, W.E., Swanger-Gagne, M., Girio, E.L., Pirvics, L., Herbst, L., Guzzo, J.L., Phillips, C., & O'Connor, B. (2009). Enhancing traditional behavioral parent training for single-mothers of children with ADHD. *Journal of Clinical Child and Adolescent Psychology*, 38, 206-218.

Chacko, A., Uderman, J.*, & Zwilling, A.* (2013). Lessons learned in enhancing behavioral parent training for high-risk families of youth with ADHD. *The ADHD Report*, 6-16.

We thank the reviewer for these useful references and we have updated the text in the manuscript accordingly (see page 8):

"Indeed, research suggests that single parents benefit more from enhanced BPT programs, suited to their specific needs (STEPP; [41,42]), compared to standard BPT programs."

6) Page 5, line 48-52 states "Parents are taught to express positive emotions (e.g., love, affection and warmth) and to inhibit expressions of negative and unsupportive emotions (e.g., anger, frustration and irritability) [16]." Is it more accurate to say that parents are to express positive behaviors and inhibit negative behaviors? You may be irritated (that's a feeling/emotion) but it's the behavioral expression that is problematic. That is true for positive emotions- these are likely better framed as behaviors to exhibit (from a BPT perspective).

We agree with the reviewer that parenting behaviors, and not parental affect, are the targets of BPT. However, an integral part of positive parenting behaviors is the adequate display of affect, which is often a separate target in traditional BPT. In order to make the text in the manuscript clearer, we have adopted the suggestions of the reviewer. See page 5:

“BPT aims to increase the demonstration of positive parenting behaviors and to inhibit negative parenting behaviors”

And (also page 5)

“Positive parenting behaviors consist of providing praise, encouragement, effective communication, demonstrating positive affect and offering physical affirmations [10,11], whereas negative parenting behaviors are described as providing inconsistent discipline, verbal criticism, corporal punishment, poor supervision, and demonstrating negative affect [12–14].”

7) The authors should review the Colalillo and Johnston meta-analysis as its relevant to these issues of the proposed work:

Colalillo, S., & Johnston, C. (2016). Parenting Cognition and Affective Outcomes Following Parent Management Training: A Systematic Review. *Clin Child Fam Psychol Rev*, 19(3), 216-235. doi:10.1007/s10567-016-0208-z

We thank the reviewer for this reference and have added this review to the introduction section of our manuscript, see page 6:

“When assessing parenting outcomes of BPT, it is important to consider a range of outcomes, particularly positive and negative parenting, management of affect, parenting stress and parenting self-efficacy. However, treatment trials of BPT have not uniformly assessed all parenting outcomes that are explicit targets of BPT [3,6], although BPT outcomes may differ for these different domains. For example, a recent systematic review investigated multiple parenting outcomes (parental affect, parenting stress, and parenting self-efficacy) of BPT, and showed positive results for outcomes most closest to the target of BPT (e.g., parenting self-efficacy), but less so for more distal outcomes (e.g., parental psychopathology) [20]. Notably, studies in that review were not specifically conducted in ADHD samples, nor were outcomes quantified using a (individual participant data) meta-analytic approach, which will be the analytic approach in the current study.”

Reviewer: 2

Reviewer Name: Nicole K Schatz

Institution and Country: Florida International University

Please state any competing interests or state ‘None declared’: None declared

Please leave your comments for the authors below

Overall, this protocol is clear and methods appear consistent with best practices for meta-analyses. A minor note, there are a couple areas that might benefit from additional explanation.

We thank the reviewer for the compliments on the consistency and clarity of our protocol.

1. The first is with regard to socio-economic status. SES is reported across studies using a variety of metrics. It would be helpful if the authors could provide additional description of how SES will be defined for the purposes of this meta-analysis.

It is indeed difficult to consistently assess SES across studies when different metrics are used. This is especially the case for income levels. We will therefore use education levels of both parents combined as a measure of SES, and categorize them as low, medium, or high. This is a relatively straightforward method, less reliant on metrics, which allows us to apply the same method consistently to almost all studies. Given that this approach is a common and acceptable method used in health research [57], we have also adopted this in our previous IPDMA [5]. We have added this to the manuscript, see page 12:

“SES will be conceptualized in line with previous work of our group and following recommendations in health research [5,57], as low (<high school), medium (high school graduate or college education), or high (>college graduate).”

2. On page 14, the authors provide helpful description of how multiple measures of the same concept will be addressed in the analyses. It would be helpful if this explanation could be expanded to include a description of how multiple raters will be addressed. For example, how will the authors address studies in which ratings are provided by both mothers and fathers?

We agree this is important to expand upon. For all measures in the method section, we will indicate which rater was used, such as parent-, child- or observer-rated. However, it can be the case that both the mother and father filled out the same assessment. If we have to choose, we will select the mother rated assessment. The reason for this is two-fold: first, mothers more often fill out the assessment (given that mothers usually take part in the training), both within and across studies, and second, mothers are more often primary caregivers, and thus have more knowledge on their child’s behavior. We have updated the manuscript accordingly, see page 12:

“If multiple raters (mothers and fathers) are included to assess the same concept, the assessment by the mother will be preferred as they are more often the primary caregiver and more often take part in BPT.”

VERSION 2 – REVIEW

REVIEWER	Anil Chacko New York University, USA
REVIEW RETURNED	14-Jul-2020

GENERAL COMMENTS	I think the authors have done a very fine job and have been responsive to addressing my concerns with their revised manuscript
--

REVIEWER	Nicole Schatz Florida International University United States
REVIEW RETURNED	16-Jul-2020

GENERAL COMMENTS	The authors have addressed my question about the treatment of studies with eligible ratings from multiple raters. With regard to my question about SES, the authors have provided a reasonable solution; however, it would be helpful for the authors to provide in their introduction their rationale for selecting parental education as their measure of SES. Additionally, the authors should acknowledge the limitations of this measure as a stand-alone indicator of SES. See below for a couple suggested citations:
---

	Braveman, P.A., Cubbin, C., Egerter, S., Chideya, S., Marchi, K.S., Metzler, M., & Posner, S. (2005). Socioeconomic status in health research. JAMA, 294(22), 2879-2888. doi:10.1001/jama.294.22.2879 Shavers, V. (2007). Measurement of socioeconomic status in health disparities research. Journal of the National Medical Association, 99(9), 1013-1023
--	---

VERSION 2 – AUTHOR RESPONSE

Reviewer: 2

Reviewer Name: Nicole Schatz

Institution and Country:

Florida International University

United States

Competing interests: None declared

With regard to my question about SES, the authors have provided a reasonable solution; however, it would be helpful for the authors to provide in their introduction their rationale for selecting parental education as their measure of SES. Additionally, the authors should acknowledge the limitations of this measure as a stand-alone indicator of SES. See below for a couple suggested citations:

Braveman, P.A., Cubbin, C., Egerter, S., Chideya, S., Marchi, K.S., Metzler, M., & Posner, S. (2005). Socioeconomic status in health research. *JAMA*, 294(22), 2879-2888. doi:10.1001/jama.294.22.2879

Shavers, V. (2007). Measurement of socioeconomic status in health disparities research. *Journal of the National Medical Association*, 99(9), 1013-1023

Reply: We thank the reviewer for these suggestions, which touch an important issue. We agree that there are limitations to using parental educational level as measure of SES. Unfortunately, in this type of study (IPDMA), we are reliant on the measures that are consistently used across studies, forcing us to make pragmatic choices at times. As far as we can see in the studies that fulfill our inclusion criteria, there are limited SES measures, mostly using parental education as proxy, with little data on income, occupation, or neighborhood. We made a small adjustment in the introduction (p.8), method (p.12) and have added the limitations of our SES measure to our discussion section (p.14). We have incorporated this into our manuscript, which now reads:

Introduction:

“Second, low socio-economic status (SES) may moderate outcomes of BPT, as it may increase family strain and impact the availability of family resources. There is mixed evidence of the moderating effect of low SES on the effects of BPT on ADHD symptoms and related problems, showing either no effect [24], or better outcomes for parents with lower SES [37]. So far, effects of low SES in relation to parenting outcomes of BPT for children with ADHD have not been investigated. For the present IPDMA we will rely on an imperfect measure of SES, i.e., parental education, as we are reliant on the measures that are consistently used across studies”

Methods:

“SES can be assessed in different ways (e.g. based on income, occupation, neighborhood or education) and due to our IPDMA methodology we are reliant on choosing the assessment measure that is most consistently used across studies. Given that data on parental education is often available, and is deemed acceptable as a proxy for SES [5,57], parental education level will serve as a proxy

measure for SES in the current study. SES will be conceptualized as low (<high school), medium (high school graduate or college education), or high (>college graduate).”

Discussion:

“Since parental education levels are easy to collect, they are omnipresent in the data. Ideally, several aspects of SES are taken into account, such as income, occupation, and/or neighborhood [38,65], but this data is available in only few datasets.”

VERSION 3 – REVIEW

REVIEWER	Nicole Schatz Florida International University, United States
REVIEW RETURNED	09-Sep-2020
GENERAL COMMENTS	Thank you for the opportunity to review this manuscript. The authors have addressed my prior comments.